# Childhood overweight and obesity at the start of primary school: External validation of pregnancy and early-life prediction models

Nida Ziauddeen[1,2]*, Paul J. Roderick[1], Gillian Santorelli[3], John Wright[3], Nisreen A. Alwan[1,2,4]*

1 School of Primary Care, Population Sciences and Medical Education, Faculty of Medicine, University of Southampton, Southampton, United Kingdom, 2 NIHR Applied Research Collaboration Wessex, Southampton, United Kingdom, 3 Bradford Institute for Health Research, Bradford Royal Infirmary, Bradford, United Kingdom, 4 NIHR Southampton Biomedical Research Centre, University of Southampton and University Hospital Southampton NHS Foundation Trust, Southampton, United Kingdom

* Nida.Ziauddeen@soton.ac.uk (NZ); N.A.Alwan@soton.ac.uk (NAA)

**Data Availability Statement:** Data requests can be made directly to Born in Bradford by completing an expression of interest form available from https://borninbradford.nhs.uk/research/how-to-access-

## Abstract

Tackling the childhood obesity epidemic can potentially be facilitated by risk-stratifying families at an early-stage to receive prevention interventions and extra support. Using data from the Born in Bradford (BiB) cohort, this analysis aimed to externally validate prediction models for childhood overweight and obesity developed as part of the Studying Lifecourse Obesity PrEdictors (SLOPE) study in Hampshire. BiB is a longitudinal multi-ethnic birth cohort study which recruited women at around 28 weeks gestation between 2007 and 2010 in Bradford. The outcome was body mass index (BMI) $\geq$91st centile for overweight/obesity at 4–5 years. Discrimination was assessed using the area under the receiver operating curve (AUC). Calibration was assessed for each tenth of predicted risk by calculating the ratio of predicted to observed risk and plotting observed proportions versus predicted probabilities. Data were available for 8003 children. The AUC on external validation was comparable to that on development at all stages (early pregnancy, birth, ~1 year and ~2 years). The AUC on external validation ranged between 0.64 (95% confidence interval (CI) 0.62 to 0.66) at early pregnancy and 0.82 (95% CI 0.81 to 0.84) at ~2 years compared to 0.66 (95% CI 0.65 to 0.67) and 0.83 (95% CI 0.82 to 0.84) on model development in SLOPE. Calibration was better in the later model stages (early life ~1 year and ~2 years). The SLOPE models developed for predicting childhood overweight and obesity risk performed well on external validation in a UK birth cohort with a different geographical location and ethnic composition.

## Introduction

Childhood overweight and obesity has been identified as one of the most serious public health challenges of the 21st century with 38 million children aged under 5 years and over 340 million aged 5 to 19 years overweight or obese globally [1]. The first '1000' days, the time from conception to age 2 years, is recognised to be a critical period of development and is also a period of

data/ and submitting it to the BiB Programme Director (rosie.mceachan@bthft.nhs.uk).

**Funding:** This analysis received no specific grant from any funding agency, commercial or not-for-profit sectors. Born in Bradford (BiB) receives core infrastructure funding from the Wellcome Trust (WT101597MA), and a joint grant from the UK Medical Research Council (MRC) and UK Economic and Social Science Research Council (ESRC) (MR/N024397/1) and one from the British Heart Foundation (BHF) (CS/16/4/32482). The National Institute for Health Research Yorkshire and Humber ARC, and Clinical Research Network both provide support for BiB research. SLOPE was supported by an Academy of Medical Sciences and Wellcome Trust Grant [AMS_HOP001\1060].

**Competing interests:** The authors have declared that no competing interests exist.

more intensive contact with health care professionals. Utilising this close contact to risk-stratify families at an early-stage to receive prevention interventions and extra support could be one approach to tackling the childhood obesity epidemic. Weight can be a sensitive topic to raise [2] and our consultation work with practitioners suggests that health professionals would like an 'objective' way to stratify risk rather than individualised clinical judgement which feels subjective. A meta-analysis of 48 studies concluded that there was a high degree of BMI tracking over time and a low probability of weight change without weight loss treatment [3]. A five year longitudinal study of 5863 students aged 11–12 years at baseline recruited from 36 London schools found little evidence of new cases of overweight and obesity emerging over adolescence but few overweight and obese adolescents reduced to a healthy weight [4] supporting the case for targeting obesity prevention strategies in early years. National guidance for the clinical management of overweight and obesity in both adults and children in the UK recommends multicomponent interventions which include behaviour change strategies to increase physical activity levels/decrease inactivity, improve eating behaviour and diet quality, and reduce energy intake [5]. Existing guidance on management of overweight and obesity could be adapted to target prevention in high-risk groups.

Prediction models are used to estimate the probability of developing a particular disease or outcome. Prediction models can provide more accurate risk estimates compared to more subjective predictions [6] and can augment clinical judgement [7] to enable intervention at an early stage before the development of the disease or outcome under consideration. Prediction models generated using routinely collected data are easier to apply in clinical practice as they only utilise data that is already being collected. A systematic review identified eight prediction models for the risk of childhood overweight and obesity [8]. The age at which outcome was predicted in the models varied from 1 to 13 years making it difficult to combine or compare models against each other. Only four of the eight prediction scores were externally validated. Inappropriate handling of missing data and discarding information through categorisation of continuous variable during model development can introduce bias. Two studies reported carrying out multiple imputation to handle missing data, the other studies either carried out complete case analysis or did not report presence/handling of missing data. Four studies retained all continuous predictors as continuous whereas the other studies categorised or dichotomised some or all continuous variables. Maternal pre-pregnancy BMI, infant gender and birthweight were the most common predictors but no single risk factor was included in all the prediction models. Model discrimination using area under the curve (AUC) ranged from 0.64 to 0.91 but only two could be applied in routine healthcare in the UK as predictors related to the father (such as paternal BMI or employment) or household (such as parental education, smoking in the household, number of siblings, income) are not routinely collected and may be complex to measure routinely. The two models which could be applied in routine healthcare both incorporated weight z-scores calculated using UK 1990 reference values [9].

As part of the Studying Lifecourse Obesity PrEdictors (SLOPE) study, we utilised anonymised routinely-collected antenatal and birth records linked to child health records for births registered at University Hospital Southampton (UHS) in the South of England between 2003 and 2018 to develop childhood overweight and obesity prediction models. UHS provides maternity care to residents in the city of Southampton and the surrounding areas of Hampshire. All maternal and birth variables in SLOPE were collected as part of healthcare for pregnant women in the study region. As part of England's National Child Measurement Programme (NCMP), childhood weight and height were measured in 30,958 children at 4–5 years which was defined as the sample for the developing the prediction model. The models were developed in stages, incorporating data collected at first antenatal appointment (booking), later pregnancy/birth, and early-life predictors (~1 and ~2 years). Logistic regression

with backward stepwise elimination and fractional polynomials were used to develop the models. Models were internally validated using bootstrapping (1000 repetitions). Models were well calibrated and area under the curve improved from 0.64 for the model only incorporating maternal predictors to 0.82 when incorporating all predictors up to child weight at ~2 years [10].

As a model usually performs better in the data used for its development, external validation in data that were not used to develop the model is needed to quantify model performance and to assess generalizability before application in practice. Model performance is evaluated using discrimination and calibration. This analysis aimed to externally validate the SLOPE prediction models using outcome data from children aged 4–5 years from the Born in Bradford (BiB) cohort [11].

## Methods

Data from the Born in Bradford (BiB) cohort was used for the external validation of the SLOPE models. BiB is a longitudinal multi-ethnic birth cohort study which recruited 12,453 women comprising 13,776 pregnancies at around 28 weeks gestation between 2007 and 2010 in Bradford, located in the North of England. Written informed consent for the data collection and linkage to routine data was provided by the child's caregiver. Ethics approval was granted by the National Health Service Health Research Authority Yorkshire and the Humber (Bradford Leeds) Research Ethics Committee (reference: 16/YH/0320).

### Outcome

As part of the National Child Measurement Programme (NCMP) [12], height and weight are measured in all children attending state schools in England at 4–5 and 10–11 years. This coincides with the first and final years of primary school in England. Parental consent for linkage to the BiB cohort was provided at recruitment, but parents can opt-out of consenting to NCMP measurements. BMI was converted to age- and sex-adjusted BMI z-scores according to the UK 1990 growth reference charts [9, 13]. The outcome was overweight/obesity at 4–5 years, and was defined as BMI $\geq$91st centile based on the UK clinical cut-off [5, 14] used to develop the SLOPE models.

### Predictors from the SLOPE models

Maternal height (cm), self-reported ethnicity, education and smoking during pregnancy were obtained from an administered baseline questionnaire which was completed at recruitment at approximately 26–28 weeks gestation. Detailed self-reported ethnicity was condensed to White, Mixed, Asian, Black/African/Caribbean and Other. Highest maternal educational qualification was categorised as secondary (GCSE) and under, college (A levels), and university degree or above. Smoking was categorised as current smoker, ex-smoker, or non-smoker. Maternal weight at first antenatal booking appointment (approximately 12 weeks gestation), gestational age at birth, birthweight and child sex were obtained from electronic maternity records. Maternal BMI was calculated using weight measured at pregnancy booking and height from the baseline questionnaire. With regards to child BMI in early life, weight and height was measured at 12 and 24 months of age in a sub-cohort of the total BiB sample (BiB 1000) [15]. Participants in BiB 1000 were recruited as part of an intensive follow-up during infancy to study the patterns and aetiology of childhood obesity. All measurements were taken by researchers who received training in anthropometric measurements at the beginning of BiB using a study measurement protocol/standard. Where infant anthropometric measurements

were missing, they were supplemented using linked data collected by health visitors as part of routine NHS care in the UK.

The four model stages were first antenatal appointment (booking), birth, early life ~1 year and early life ~2 years. Maternal BMI at booking, smoking status, ethnicity, intake of folic acid supplements and partnership status were predictors at every stage. Additional predictors at each model stage were (1) maternal age, first language and parity (booking); (2) maternal age, educational attainment, first language, parity, birthweight and gestational age at birth (birth); (3) maternal age, educational attainment, birthweight, gestational age at birth, infant gender and weight at ~1 year (early life ~1 year); and (4) educational attainment, birthweight, gestational age at birth, infant gender and weight ~2 years (early life ~2 years).

## Statistical analysis

All analysis was conducted using Stata 15 [16]. The sample was restricted to singleton children with a valid height and weight measurement at 4–5 years (to calculate the outcome of interest). Estimated risk of overweight and obesity at 4–5 years was calculated using the SLOPE CORE tool. Missing predictor values were imputed using multiple imputation by chained equations (MICE) with truncated regression for continuous variables and predictive mean matching for categorical variables. The percentage of missing data was highest for early life weight at ~1 and ~2 years and this was used to decide the number of imputations (55 imputations with 10 iterations per imputed dataset) based on the recommendation that the number of imputations equals the percentage of missing data in the dataset [17]. The results from analyses of each of the imputed datasets were combined to produce estimates and confidence intervals that incorporate the uncertainty of imputed values.

Predictive performance was assessed by examining discrimination and calibration. Discrimination is a measure of how well the model differentiates between individuals and was quantified by calculating the area under the receiver operating curve (AUC). An AUC value of 0.5 represents no discrimination capacity, with discrimination improving up to the value of 1 which represents perfect discrimination. Calibration measures how well the predicted outcome of the model agrees with the observed outcome on average [18]. This was assessed for each tenth of predicted risk, ensuring 10 equally sized groups, by calculating the ratio of predicted to observed risk and plotting observed proportions versus predicted probabilities.

A risk prediction tool is used to identify high risk groups by classifying individuals into high and low risk groups using a pre-defined risk threshold. A risk threshold of 20% was reached for the SLOPE CORE (Childhood Obesity Risk Estimation) tool which was guided by prediction performance, local stakeholder consultation and prediction tools for other outcomes used in UK healthcare [10]. Thus, we compared the proportion identified using risk thresholds of 20%, 25% and 30% to that subsequently developed overweight and obesity at 4–5 years.

## Results

The BiB cohort included 8003 eligible children (Fig 1). Complete data for all predictors was available for 22.3% of women/children (n = 1824). Most participants were missing data on none (n = 1824, 22.8%) or one (n = 2889, 36.1%) predictor. The percentage of missing data was highest for early life weight at ~1 (54.4%) and ~2 years (41.4%). The percentage of missing data for other predictors ranged from 17.8 to 21.4%.

Cohort characteristics are presented in Table 1. Mean maternal age at booking was 27.3 years (standard deviation (SD) 5.5). Mean maternal BMI at booking was 26.0 kg/m$^2$ (SD 5.6). Nearly three-quarters of women reported being never smokers. Nineteen percent of the

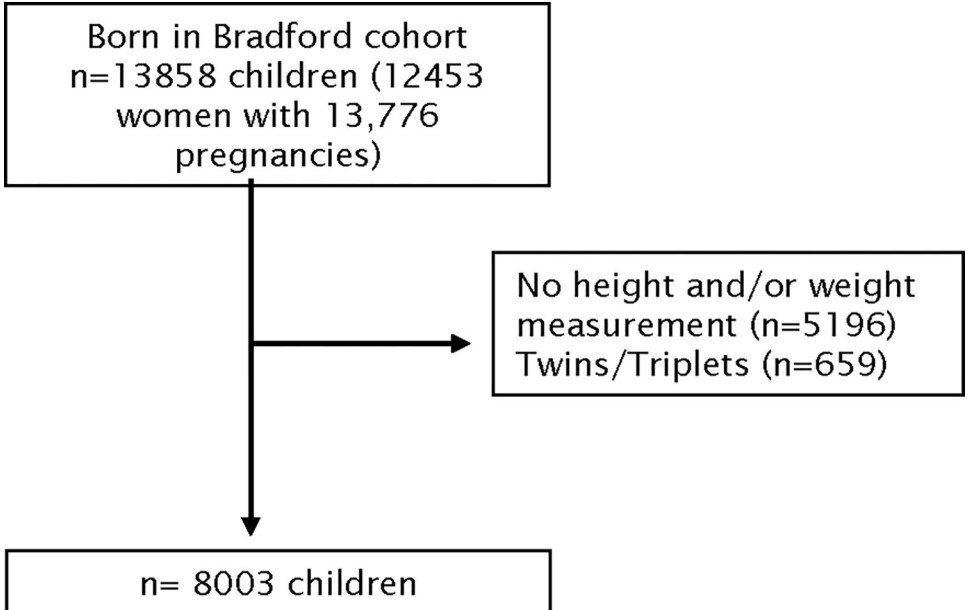

**Fig 1. Flow diagram showing the eligible sample (n = 8003) for this analysis after excluding children without a valid weight and/or height measurement and twins/triplets.**

women had a university degree or a higher qualification. The cohort was predominantly of Asian (60.3%) or White (36.0%) ethnicity. Two-thirds of women reported English as a second language. Fifteen percent of mothers reported being a lone parent at the booking appointment. The prevalence of overweight and obesity at 4–5 years was 14.5%. The prevalence of overweight and obesity was higher in children of mothers who smoked during pregnancy (17.2%) than children of mothers who were never smokers (14.0%).

The AUC on external validation was comparable to that at model development in SLOPE at all stages (booking, birth, ~1 year and ~2 years) (Table 2). The AUC at development was 0.66 (95% confidence intervals (CI) 0.65 to 0.67) at booking compared to 0.64 (95% CI 0.62 to 0.66) on external validation. Similarly, the AUC was 0.83 (95% CI 0.82 to 0.84) at ~2 years at development and 0.82 (95% CI 0.81 to 0.84) on external validation.

Fig 2 shows the agreement between mean observed and mean predicted risk grouped by tenths of predicted risk for the four model stages. The SLOPE model at later stages (~1 year and ~2 years) gave a more accurate estimate of predicted risk. There was less agreement between observed and predicted risk grouped by tenth of risk at the early stages (booking and birth) in individuals with higher risk indicative of overestimation of risk.

Using a 20% risk threshold, 36.9% were identified as high risk at booking and captured 53.3% of overweight and obesity events at 4–5 years (Table 3). The proportion identified as high risk decreased to 21.2% at ~2 years and captured 59.6% of events. Using higher risk thresholds (25% and 30%) identified fewer children as high risk but also captured fewer events. This indicates that the optimal threshold for determining high risk using these models is 20%.

## Discussion

We evaluated the performance of the SLOPE models using data from the South of England for predicting the risk of overweight and obesity at 4–5 years in the BiB cohort in the north of England. Discrimination of the SLOPE equations in this cohort was comparable to that in the

**Table 1. Descriptive statistics and percentage overweight or obese by predictor at age 4–5 years.**

| Predictors | Mean ± SD | % of children ≥91st centile BMI |
|---|---|---|
| N | 8003 | |
| Maternal age at booking, years | 27.3 ± 5.6 | - |
| Maternal BMI at booking, kg/m$^2$ | 26.0 ± 5.6 | - |
| Birthweight, kg | 3.2 ± 0.6 | - |
| Gestational age at birth, days | 276 ± 13 | - |
| Child weight at ~1 year, kg | 9.1 ± 1.2 | - |
| Child weight at ~2 years, kg | 12.6 ± 1.6 | - |
| | **% (95% CI)** | |
| Maternal smoking status | | |
| Never smoked | 73.4 | 14.0 |
| | 72.4 to 74.4 | |
| Ex-smoker | 12.9 | 14.3 |
| | 12.1 to 13.6 | |
| Current smoker | 13.7 | 17.2 |
| | 12.9 to 14.5 | |
| Maternal educational attainment | | |
| University or above | 19.2 | 12.3 |
| | 18.4 to 20.1 | |
| College (A levels) | 11.3 | 13.1 |
| | 10.6 to 12 | |
| Secondary (GCSE) or lower | 69.4 | 15.3 |
| | 68.4 to 70.5 | |
| Maternal ethnicity | | |
| White | 36.0 | 15.5 |
| | 34.9 to 37.1 | |
| Mixed | 1.7 | 13.5 |
| | 1.4 to 2.0 | |
| Asian | 60.3 | 13.7 |
| | 59.1 to 61.4 | |
| Black/African/Caribbean | 1.4 | 19.6 |
| | 1.1 to 1.7 | |
| Other | 0.7 | 17.8 |
| | 0.5 to 0.8 | |
| Maternal intake of folic acid supplements | | |
| Started taking once pregnant | 78.8 | 14.5 |
| | 77.8 to 79.8 | |
| Not taking supplement | 21.2 | 14.3 |
| | 20.2 to 22.2 | |
| Maternal first language English | | |
| No | 33.9 | 14.1 |
| | 32.9 to 34.9 | |
| Yes | 66.1 | 14.7 |
| | 65.1 to 67.1 | |
| Partnership status at booking | | |
| Partnered | 85.3 | 14.2 |
| | 84.4 to 86.1 | |

*(Continued)*

**Table 1.** (Continued)

| Predictors | Mean ± SD | % of children ≥91st centile BMI |
|---|---|---|
| Single | 14.7 | 15.6 |
| | 13.9 to 15.6 | |
| Parity at booking | | |
| 0 | 36.8 | 14.4 |
| | 35.7 to 37.8 | |
| 1 | 27.9 | 14.1 |
| | 26.9 to 28.8 | |
| 2 | 17.3 | 13.3 |
| | 16.5 to 18.1 | |
| ≥3 | 18.0 | 16.2 |
| | 17.2 to 18.9 | |
| Child sex | | |
| Male | 51.3 | 15.0 |
| | 50.2 to 52.4 | |
| Female | 48.7 | 13.9 |
| | 47.6 to 49.8 | |
| Overweight/obese at 4–5 years | | |
| No | 85.5 | - |
| | 84.8 to 86.3 | |
| Yes | 14.5 | - |
| | 13.7 to 15.2 | |

development cohort at all model stages. Risk was over-predicted (predicted risk was higher than observed risk) in higher risk groups using the early pregnancy and birth equations but calibration was good in the later model stages (early life ~1 year and ~2 years). Using a 20% risk threshold, 36.9% were identified as high risk at booking and captured 53.3% of overweight and obesity events at 4–5 years.

Of eight prediction models for childhood overweight and obesity previously identified in a systematic review, four have been externally validated [8]. Prediction models for childhood obesity developed using data from the 1986 Northern Finland Birth Cohort were externally validated in a retrospective cohort in Italy and a prospective birth cohort (Project Viva) in the US. The AUC was slightly lower in the US external validation cohort than the Finnish development cohort (0.73 vs 0.78) but calibration was not satisfactory. A modified version of the model excluding two predictors was externally validated in the Italian cohort and the AUC was 0.73 on development and 0.70 on external validation with adequate calibration [19].

In the UK, a risk prediction algorithm developed using data from the UK Millennium Cohort Study (MCS) [20] was externally validated in a 10% sample of the Avon Longitudinal Study of Parents and Children (ALSPAC) known as the Children in Focus (CiF). Of the 1432 children in this sub-sample, data on child weight was available for 980 children and this was the final sample for external validation. The AUC on developing the model in MCS was 0.72. Applying the model in the ALSPAC validation sample resulted in an AUC of 0.67. The AUC increased to 0.70 on model recalibration [21].

The prediction model developed using data from the BiB cohort was also validated using the CiF subsample of ALSPAC. Models were developed for use at three stages (6±1.5 months, 9±1.5 months and 12±1.5 months). The AUC ranged from 0.86 to 0.91 on model development, and from 0.85 to 0.89 on external validation [22].

**Table 2. Discrimination and calibration performance for the SLOPE models in estimating risk of overweight and obesity at 4–5 years in the BiB cohort using predictor data at booking, birth and early life (~ 1 and 2 years).**

| | Booking | Birth | Early life (~1 year) | Early life (~2 years) |
|---|---|---|---|---|
| **Discrimination (AUC, 95% CI)** | | | | |
| SLOPE development | 0.66 | 0.69 | 0.78 | 0.83 |
| | 0.65 to 0.67 | 0.68 to 0.70 | 0.77 to 0.79 | 0.82 to 0.84 |
| BiB external validation | 0.64 | 0.65 | 0.75 | 0.82 |
| | 0.62 to 0.66 | 0.64 to 0.67 | 0.74 to 0.77 | 0.81 to 0.84 |

In most externally validated models, AUCs on external validation were slightly lower than in development models, and this in line with our study. An exception is a risk index which was developed using data from a retrospective cross-sectional school based cohort collected in 2007 in Greece which had a similar AUC of 0.64 on development and external validation in a cross-sectional school based cohort collected in 2012–13 [23].

In terms of assessing calibration, one published study reported model calibration [19] but used a different measure (Hosmer-Lemeshow test) to the measure used in our study. It reported unsatisfactory calibration on external validation in one of the two cohorts they used, while our models remained well calibrated on external validation.

## Strengths and limitations

The SLOPE models were developed in a population-based cohort in the South of England using routinely collected healthcare data. The prevalence of the outcome was comparable in both the development (14.8%) and external validation (14.5%) cohorts and was measured as part of the NCMP thus limiting variation in measurement practices. Key differences between the development and external validation cohorts were the ethnic composition where the development cohort was predominantly White (90%) whereas the external validation cohort was 60% South Asian and 31% White. This is reflected in the slightly lower average birthweight [24] in the overall sample and a higher proportion reporting English as a second language.

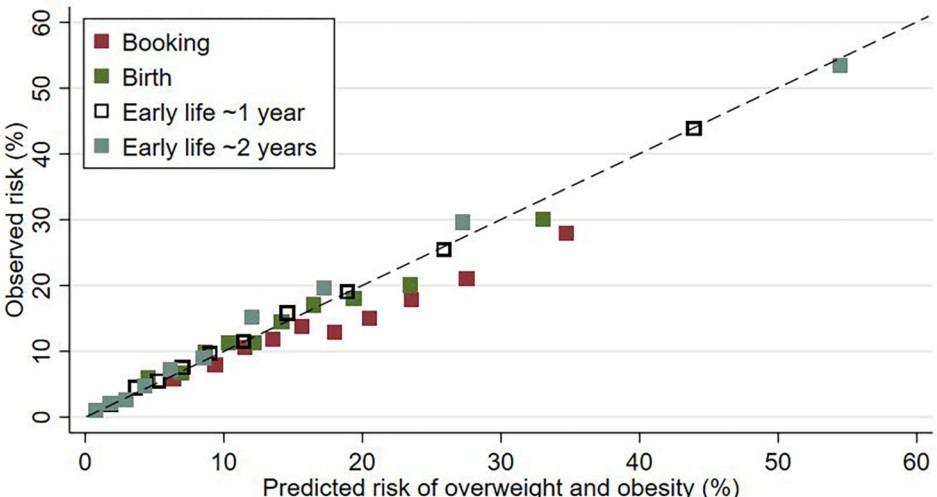

**Fig 2. Predicted versus observed risk of overweight and obesity at 4–5 years by model stage for the SLOPE CORE tool by tenth of risk.**

**Table 3. Proportion (95% CI) of overweight and obesity in children at 4–5 years classified as low and high risk by model stage using predictor data at booking, birth and early life (~ 1 and 2 years).**

| | Proportion identified as high risk | Low risk | | High risk | |
|---|---|---|---|---|---|
| | | Not overweight and obese | Overweight and obese | Not overweight and obese | Overweight and obese |
| **20% risk threshold** | | | | | |
| Booking | 36.9 | 65.9 | 46.7 | 34.1 | 53.3 |
| | | 64.6 to 67.2 | 43.5 to 50.0 | 32.8 to 35.4 | 50.0 to 56.5 |
| Birth | 23.0 | 79.6 | 61.3 | 20.4 | 38.7 |
| | | 78.6 to 80.7 | 58.2 to 64.5 | 19.3 to 21.5 | 35.5 to 41.8 |
| Early life (~1 year) | 22.7 | 82.2 | 48.0 | 17.8 | 52.0 |
| | | 81.0 to 83.4 | 4.5 to 51.4 | 16.6 to 19.0 | 48.6 to 55.5 |
| Early life (~2 years) | 21.2 | 85.3 | 40.4 | 14.7 | 59.6 |
| | | 84.3 to 86.3 | 36.9 to 43.9 | 13.7 to 15.7 | 56.1 to 63.1 |
| **25% risk threshold** | | | | | |
| Booking | 20.9 | 81.5 | 65.1 | 18.5 | 34.9 |
| | | 80.4 to 82.6 | 62.0 to 68.3 | 17.4 to 19.6 | 31.7 to 38.0 |
| Birth | 11.9 | 90.1 | 76.4 | 9.9 | 23.6 |
| | | 89.3 to 90.9 | 73.7 to 79.1 | 9.1 to 10.7 | 20.9 to 26.3 |
| Early life (~1 year) | 15.8 | 88.5 | 59.2 | 11.5 | 40.8 |
| | | 87.5 to 89.5 | 55.9 to 62.5 | 10.5 to 12.5 | 37.5 to 44.1 |
| Early life (~2 years) | 16.3 | 89.6 | 49.0 | 10.4 | 50.1 |
| | | 88.7 to 90.4 | 45.5 to 52.6 | 9.6 to 11.3 | 47.4 to 54.5 |
| **30% risk threshold** | | | | | |
| Booking | 10.0 | 91.6 | 80.9 | 8.4 | 19.1 |
| | | 90.7 to 92.4 | 78.3 to 83.4 | 7.6 to 9.3 | 16.6 to 21.7 |
| Birth | 5.9 | 95.6 | 85.9 | 4.4 | 14.1 |
| | | 95.0 to 96.1 | 83.7 to 88.0 | 3.9 to 5.0 | 12.0 to 16.3 |
| Early life (~1 year) | 10.7 | 92.9 | 68.1 | 7.1 | 31.9 |
| | | 92.1 to 93.7 | 64.8 to 71.4 | 6.3 to 7.9 | 28.6 to 35.2 |
| Early life (~2 years) | 12.8 | 92.4 | 56.2 | 7.6 | 43.8 |
| | | 91.6 to 93.2 | 52.8 to 59.7 | 6.8 to 8.4 | 40.3 to 47.2 |

There was also a higher proportion of lone mothers, mothers with lower educational attainment and higher order pregnancies in BiB.

Bradford ranked 19th and Southampton 54th out of 317 local authorities in England (1 is most deprived) in 2015 so are both relatively deprived cities [25]. The use of data from more deprived areas is a strength of this analysis given the higher prevalence of overweight and obesity in more deprived areas [26]. The BiB cohort is representative of the local population in Bradford and has similarities with other UK cities with high levels of ethnic minority groups but is not representative of the rest of the country [11].

The differences between the cohorts are a strength of the analysis as it assesses model performance in a population with different characteristics and therefore the findings would be more generalisable. The performance of the SLOPE model on external validation in BiB was comparable to model performance on development and supports its use to predict risk of childhood overweight and obesity within a wider UK setting.

There was a high level of missing data for early life predictors (weight at ~1 year and ~2 years) in both the development and validation cohort. Multiple imputation was used to address this. Measuring health and weight in children at these ages is part of statutory care and so the missing data could reflect an issue in how these measurements are recorded. It is possible that

the child's handheld record is updated but not their electronic record or that the measurement is entered in open text boxes in the electronic records rather in designated response boxes which makes it difficult to access for research purposes. Recording of key variables in electronic records have improved over time and considering that the model performance was highest for age groups which also had the highest percentage of missing data, the implementation of this tool in practice may have implications for both data recording and utilisation of the tool.

BMI is a useful population-level measure of overweight and obesity as it is easy to measure and the same regardless of gender or age (once adulthood is attained). BMI is the most commonly used marker of overweight and obesity [27] and is the marker used in the NCMP in England after adjusting for age and sex. However, BMI does not account for differences in body composition and thus may not be the best marker of body fat [28] or cardio-metabolic risk in South Asian children [29].

This study also shows that the optimal threshold for identifying high risk was 20% which is what was suggested based on the sensitivity, specificity, positive predictive values and negative predictive value of the models at development [10]. At the 20% risk threshold, 52% and 60% of cases are identified using the models at ~1 year and ~2 years respectively. This reduces to 32–50% at these stages if higher risk thresholds are used.

As the model incorporates several equations requiring background calculations, we developed a website so that the tool can be used easily. The user is required to enter the values for the continuous predictors and choose the appropriate option from the dropdown for the categorical variables. Information on all predictors is required at each stage to calculate the risk score. The risk score is categorised as low risk (0–20%, green), medium risk (20–30%, yellow) and high risk ($\geq$30%, orange). The feasibility and acceptability of using the tool in practice by midwives and health visitors, including the risk cut-offs and risk communication is being explored in Wessex [30]. We are testing the usability and feasibility of the prediction tool (SLOPE CORE (Childhood Obesity Risk Estimation)) as an aid to healthcare professionals to guide delivery of an intervention rather than just as a screening test. This is based on public involvement consultations in which mothers expressed interest in early identification of risk but stressed the need for support and advice to modify risk. The tool is envisaged to provide a prompt for the health professional to introduce the topic at an early stage and to help target extra support in resource limited setting. Additionally, we envisage health professionals will use their professional judgement and continue to provide support to groups they identify as needing additional support even if they are categorised as low risk using the tool. Next steps also include developing and validating prediction models for childhood overweight and obesity using the routinely measured outcome of BMI at age 10–11 years as this captures key development of overweight and obesity.

The SLOPE models developed for predicting childhood overweight and obesity risk demonstrated good model performance on external validation in a birth cohort with a different geographical location and ethnic composition.

## Author Contributions

**Conceptualization:** Nida Ziauddeen, Paul J. Roderick, Nisreen A. Alwan.

**Data curation:** Gillian Santorelli, John Wright.

**Formal analysis:** Nida Ziauddeen.

**Supervision:** Paul J. Roderick, Nisreen A. Alwan.

**Writing – original draft:** Nida Ziauddeen.

**Writing – review & editing:** Paul J. Roderick, Gillian Santorelli, John Wright, Nisreen A. Alwan.

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
