## [Decision Letter · Decision Letter 0]

4 Oct 2021

PGPH-D-21-00446

Childhood overweight and obesity at the start of primary school: external validation of pregnancy and early-life prediction models

Dear Dr. Nida Ziauddeen,

Thank you for submitting your manuscript to PLOS Global Public Health. After careful consideration, we feel that it has merit but does not fully meet PLOS Global Public Health’s publication criteria as it currently stands. Therefore, we invite you to submit a revised version of the manuscript that addresses the points raised during the review process.

1. We recommend expanding the description of the model with more details, as well as it's contextualization as suggested by the reviewers.

2. We believe that the quality of the work will improve with a little more insight into the strengths and limitations section.

4. We recommend to raise the issue and elaborate the practical applications of the model.

5. We also recommend the revision of the text in terms of references to the literature and supplement any gaps.

As a guide to make corrections, please use the detailed reviewers comments placed below.

We look forward to receiving your revised manuscript.

Kind regards,

Hanna Nalecz

Academic Editor

Journal Requirements:

1. Please provide additional details regarding participant consent. In the ethics statement in the Methods and online submission information, please ensure that you have specified what type you obtained (for instance, written or verbal, and if verbal, how it was documented and witnessed).

2. Please provide separate figure files in .tif or .eps format only.

3. Please update the completed 'Competing Interests' statement, including any COIs declared by your co-authors. If you have no competing interests to declare, please state "The authors have declared that no competing interests exist". Otherwise please declare all competing interests beginning with the statement "I have read the journal's policy and the authors of this manuscript have the following competing interests:"

Additional Editor Comments (if provided):

We have carefully checked the type of publication pointed out by the Reviewer #2. and due to the fact that the indicated source is an abstract of the conference report, we believe that the requirements of the journal have not been violated and the article may be further processed for publication.

Thats why the first comment of the Reviewer #2. as well as the point 1., and 2. can be ignored.

Reviewers' comments:

Reviewer's Responses to Questions

**Comments to the Author**

1. Does this manuscript meet PLOS Global Public Health’s publication criteria? Is the manuscript technically sound, and do the data support the conclusions? The manuscript must describe methodologically and ethically rigorous research with conclusions that are appropriately drawn based on the data presented.

Reviewer #1: Yes

Reviewer #2: Partly

Reviewer #3: Yes

2. Has the statistical analysis been performed appropriately and rigorously?

Reviewer #1: Yes

Reviewer #2: N/A

Reviewer #3: Yes

3. Have the authors made all data underlying the findings in their manuscript fully available (please refer to the Data Availability Statement at the start of the manuscript PDF file)?

Reviewer #1: No

Reviewer #2: Yes

Reviewer #3: No

4. Is the manuscript presented in an intelligible fashion and written in standard English?

Reviewer #1: Yes

Reviewer #2: Yes

Reviewer #3: Yes

5. Review Comments to the Author

Reviewer #1: Title

Is the model not developed for predicting childhood overweight and obesity up to the age of 2? Therefore, is it applicable to primary school children?

Abstract

Good. The abstract states the main findings. Could be aided by breaking up the abstract into introduction, methods, results, and conclusion.

Introduction

Excellent introduction to the utility of risk prediction models. Very minor suggestions:

1. Add a few sentences about the strategies that could be implemented post risk stratification. Without a way to reduce the risk in high-risk populations, there is a possibility this model could pave the way for future discrimination.

2. A reference is needed for this:

“Weight can be a sensitive topic to raise and our consultation work with practitioners suggests that health professionals would like an ‘objective’ way to stratify risk rather than individualised clinical judgement which feels subjective.”

3. A subject needs to be added to this clause:

“as only utilises data that is 63 already being collected”

Methods

Excellent methods section. No further improvements to suggest here.

I do have a question, however. By including ethnicity as one of the model predictors, is there a chance that this model will become redundant in future years as the sociocultural factors that link ethnicity to childhood BMI change? Does it also limit the model’s utility to the UK as the cultural factors surrounding ethnicity will be different in different parts of the world? The model's geographical limitation to the UK and temporal usability also applies for the choice of options for maternal educational attainment and maternal first language.

Results

Interesting results.

I note the higher prevalence of childhood overweight and obesity in the group that has been grouped by the maternal ethnicity of “Black/African/Caribbean”. Is there a risk we’re reducing very heterogenous cultures into a monolith? For examples trends such as encouraging children to finish their whole bottle of milk or propping the bottle up are only present in certain cultures within this communities that have been grouped together. This model risks stereotyping everyone from said community.

Also, I note that the model is better at correctly identifying those at low risk than those at high risk. Is this model therefore best able to direct resources to those who most need it? Or will it be used instead to prohibit access to certain groups who are deemed low risk?

Discussion

I note that in your discussion you talk about the prevalence of obesity in deprived areas. But I come back to where your model seems strongest; that is in identifying those at low risk. Therefore, what are its practical uses in these settings?

The model seems to have a much higher specificity (85.3%) than sensitivity (59.6%). Is sensitivity what we should really be striving for?

Reviewer #2: Dear Authors,

Thank you for submitting the following manuscript to PLOS Global Public Health:

Manuscript ID: PGPH-D-21-00446

Type of manuscript: Research Article

Title: Childhood overweight and obesity at the start of primary school: external validation of pregnancy and early-life prediction models

Authors: Nida Ziauddeen, Paul J. Roderick, Gillian Santorelli, John Wright, Nisreen A. Alwan.

Unfortunately, at this stage, the manuscript does not qualify for being Peer Reviewed. It seems that the authors performed considerable work in order to collect the data of so many participants; however, the analysis appears to be too simple considering the sample size. Moreover, other shortcomings need to be corrected and clarified:

1. The same research was published elsewhere. As found on the following link: https://jech.bmj.com/content/74/Suppl_1/A26.1. On the 10th of September 2020, as an oral presentation, which contradicts the PLos Journal policy. Please justify.

2. The Introduction is also a repetition of the same published paper (https://jech.bmj.com/content/74/Suppl_1/A26.1.)

3. In the results section,

- Figure 1: needs more clarification. Kindly add a caption or words to the figure to help guide the reader and ease the reading of the figure.

- Table 2 and Table 3: the title needs to be more representative of the data provided; specifically, suggesting adding “predictor” at 4-5. It is unclear because the authors reported the 1&2 years old on the table but referred to 4-5 years old in the title.

4. The Discussion section:

- The authors only reported the results, and no supporting studies were included.

- In the strengths and limitations, it is advised to be more precise in reporting the strengths and limitations of the study.

5. The conclusion of the study is unclear and needs improvement.

Suppose the authors feel that they can adequately address all the issues and considerably improve the quality of this study by performing a more in-depth analysis. In that case, they can resubmit their manuscript as a new submission.

Wish you every success if you choose to submit it elsewhere.

Good luck and best wishes

Reviewer #3: Please see attached pdf for comments

6. PLOS authors have the option to publish the peer review history of their article (what does this mean?). If published, this will include your full peer review and any attached files.

**Do you want your identity to be public for this peer review?** For information about this choice, including consent withdrawal, please see our Privacy Policy.

Reviewer #1: **Yes: **Soham Bandyopadhyay

Reviewer #2: No

Reviewer #3: No

---

## [Decision Letter · Decision Letter 1]

27 Jan 2022

PGPH-D-21-00446R1

Childhood overweight and obesity at the start of primary school: external validation of pregnancy and early-life prediction models

Dear Dr.Ziauddeen,

Thank you for submitting your manuscript to PLOS Global Public Health. After careful consideration, we feel that it has merit but does not fully meet PLOS Global Public Health’s publication criteria as it currently stands. Therefore, we invite you to submit a revised version of the manuscript that addresses the points raised during the review process.

The reviewers have found your paper to be suitable for publication in PGPH. However, reviewer 1 has raised some minor issues that will need your attention:

They felt that the utility of the model, and its limitation needs to be explained more clearly with no hesitation.

I have noticed the following issues after going through the manuscript, which must be addressed before it can be accepted for publication:

1. It appears lines 150-156 are repetitions of lines 130-137. Please, check this. You could merge some of the information in the former with the latter.

2. More importantly, the discussion section needs to be reworked on. It looks more like a results section. The detail statistics (e.g. OR, CIs) are not needed in this section. They should be removed, and the results discuss in the context of the existing literature. 

We look forward to receiving your revised manuscript.

Kind regards,

Dickson Abanimi Amugsi, PhD

Academic Editor

Journal Requirements:

1. Please amend your Financial Disclosure statement. If you did not receive any funding for this study, please simply state: “The authors received no specific funding for this work.”
---

## [Editor Report · Decision Letter 2]

9 Mar 2022

Childhood overweight and obesity at the start of primary school: external validation of pregnancy and early-life prediction models

PGPH-D-21-00446R2

Dear Dr Ziauddeen,

We are pleased to inform you that your manuscript 'Childhood overweight and obesity at the start of primary school: external validation of pregnancy and early-life prediction models' has been provisionally accepted for publication in PLOS Global Public Health.

Best regards,

Dickson Abanimi Amugsi, PhD

Academic Editor